# Dietary Heme-Containing Proteins: Structures, Applications, and Challenges

**DOI:** 10.3390/foods11223594

**Published:** 2022-11-11

**Authors:** Yilin Xing, Shanxing Gao, Xinyu Zhang, Jiachen Zang

**Affiliations:** Beijing Key Laboratory of Functional Food from Plant Resources, College of Food Science and Nutritional Engineering, China Agricultural University, Beijing 100083, China

**Keywords:** iron, heme-containing protein, hemoglobin, myoglobin, iron supplement

## Abstract

Heme-containing proteins, commonly abundant in red meat and blood, are considered promising dietary sources for iron supplementation and fortification with higher bioavailability and less side effects. As the precise structures and accurate bioactivity mechanism of various heme-containing proteins (hemoglobin, myoglobin, cytochrome, etc.) are determined, many methods have been explored for iron fortification. Based on their physicochemical and biological functions, heme-containing proteins and the hydrolyzed peptides have been also widely utilized as food ingredients and antibacterial agents in recent years. In this review, we summarized the structural characterization of hemoglobin, myoglobin, and other heme proteins in detail, and highlighted recent advances in applications of naturally occurring heme-containing proteins as dietary iron sources in the field of food science and nutrition. The regulation of absorption rate, auto-oxidation process, and dietary consumption of heme-containing proteins are then discussed. Future outlooks are also highlighted with the aim to suggest a research line to follow for further studies.

## 1. Introduction

Iron, an essential trace element for the human body, has a considerable moderating effect on human health. The bodily iron content of a 70 kg reference man is around 4.0–5.0 g. However, iron deficiency is one of the most prevalent worldwide nutritional disorders affecting about 2 billion people in the world. Anemia, the most severe stage of iron deficiency, usually leaves undesirable short-term and long-term effects on susceptible populations, especially pregnant women and infants [1,2]. From the research of Cook et. al., the reason why iron deficiency is more severe in developing countries is related closely to the diet, with less iron bioavailability and thus less iron absorption [3]. While iron supplementation is beneficial in a targeted population, iron fortification of the diet is a preferable method to prevent iron deficiency.

From a nutritional point of view, dietary iron is commonly classified into two forms: heme iron and non-heme iron. Heme is a kind of porphyrin compound which binds iron. It is the auxiliary group of heme proteins, including hemoglobin, myoglobin, cytochrome, peroxidase, catalase, etc. Hemoglobin and myoglobin are typical heme-binding proteins, which are partners in the transport and storage of oxygen in vertebrates. Hemoglobin is found packed at high concentrations (~20 mM) in red blood cells and myoglobin in aerobic muscle tissue [4]. Comparably, heme proteins are greater sources rather than non-heme proteins. We summarized the detailed information on hemoglobin, myoglobin, and other heme proteins, majored in their structural and nutritional characteristics, especially their binding ways with iron. Then, their applications in nutrition, coloring, and antibacterial potential in food supplements were mentioned. In addition, the future utilization of these proteins was also discussed to support more strategies for iron supplementation.

It is assumed that the evolution of metalloproteins is the adaption of an aerobic environment, oxidative metabolism, and prevention from harmful free radicals [5]. Studies of heme proteins and related compounds have greatly promoted our understanding of their functions such as oxygen storage and transport, degradation of reactive radicals, biological oxygenation, etc. Moreover, these studies have driven the development of production, application, and further practices of heme proteins.

## 2. Structure

### 2.1. Hemoglobin

When it comes to food-derived heme proteins, hemoglobin (Hb) is a typical one which is an essential component of blood with great significance in medical and nutritional applications [6]. However, a great amount of blood is discharged in the form of sewage, which not only causes waste of resources but also causes severe environmental pollution. For biological utilization of blood and related products, blood proteins can be separated after anticoagulation and centrifugation for further research [7]. The main component of the proteins is hemoglobin, which accounts for great significance in medical health and nutrition.

Hemoglobin has four subunits: two identical α chains and β chains combined with hemes, held together by noncovalent interactions. In Figure 1, each α subunit is paired with a β subunit identically, so that hemoglobin can be considered either a tetramer of four subunits or a dimer of αβ protomers. As further shown in Figure 1A, the α_1_β_1_ and α_2_β_2_ dimers come into contact and assume a 2-fold symmetry with the axis passing through the central cavity surrounded by four subunits. An identical heme molecule is located inside the cleft of each subunit, interacting with the hydrophobic amino acid around, such as Histidine and Leucine (Figure 1B,C). The α chains and β chains share a very high similarity of tertiary structure and the conserved domain to bind heme. As the most important role of hemoglobin is to bind and deliver oxygen in the living body, the interaction between heme and oxygen is attractive. As shown in Figure 1D, the di-oxygen molecule binds iron by an O-Fe bond. The whole structure of backbone maintains almost no change compared with reduced hemoglobin, as shown in Figure 1B,C.

Although the structure of hemoglobin has been known for almost sixty years, there are many properties related to its structure and function that are still to be further determined. To have a profound understanding of the structure of the hemoglobin(s), recent studies have found the crystal structure of a carbonmonoxy form of the hemoglobin from the fish *Eleginops maclovinus*. It had been proved that hemoglobin isolated from the Antarctic fishes may reflect a unique structure of tertiary and quaternary states, in which the distal His is displaced from its canonical position. The novel crystallographic data also represents a promising research prospect for hemoglobin functional properties as well as into the structural basis of some spectroscopic features sharing with the entire hemoglobin superfamily [10]. Moreover, the techniques of structure determination are quickly developing on the way. With the development of direct electron detectors, the usage of a Volta phase plate enables researchers to determine the structure of human hemoglobin (64 kDa) at 3.2 Å [11].

### 2.2. Myoglobin

Myoglobin (Mb), a prototypical heme-protein in the muscle of vertebrates, is the first protein to have its structure solved and its functions investigated in detail. As a simple monomeric heme-protein, myoglobin plays a significant role in tissue oxygenation, nitric oxide signal regulation, as well as bioavailable iron supplements.

Myoglobin is composed of a polypeptide chain and a heme prosthetic group, with iron porphyrin as the active center, and has a compact spherical shape. The molecule consists of eight approximately-straight segments of right-handed α-helix interrupted by bends, some of which are β-turns. As shown in Figure 2, more than 70% of residues in myoglobin are in these α-helical regions. As myoglobin has its highly ordered arrangement in crystal, the X-ray analysis has determined the accurate position of each of the R groups, which fill up nearly all the compact inner space unoccupied by backbone atoms. The positions of the side chains reflect the structure mainly stabilized by the hydrophobic effect. Most of the hydrophobic R groups are in the interior of the protein molecule, while all but two of the polar R groups are distributed on the surface of the molecule and hydrated [12]. Recently, Narita and co-workers pointed out that the two types of hydrophobic core networks which exist in myoglobin have been elucidated by continuous folding structure units [13]. Different from oxidized hemoglobin, the di-oxygen molecule interacts with iron by two O-Fe bonds, which is also fixed by the nitrogen atom from lateral histidine (Figure 2A) [14]. Interestingly, carbon monoxide has a different binding way (Figure 2B) [15], which will be discussed in the next part.

Last year, an effective theory approach was developed to investigate the phase properties of globin, which builds directly on the tertiary structure of proteins. Based on the detailed three-dimensional structure, this theoretical approach has successfully described the whole process of how myoglobin unfolds into a random coil when temperature and acidity increase [16]. Moreover, scientists have designed a series of catalysis with the knowledge of the exact structure of myoglobin, such as the double mutant F43Y/F46S Mb (high-efficiency peroxidase), F43Y/F138W/P88W Mb (high-efficiency bioremediation enzyme), F43Y/H64D Mb (an effective artificial dehaloperoxidase), etc. [17,18,19]. Based on the fact that the detailed structure of myoglobin structure has been determined a long time before, researchers can further the study of myoglobin’s properties and applications in many aspects.

### 2.3. The Allosteric Effect

In heme containing proteins, the allosteric effect refers to the phenomenon that multi-subunit proteins change their conformation and activity due to the binding of effectors. It is the crucial ability of the two heme proteins that they discriminate selectively between two chemically and structurally similar molecules, namely oxygen and carbon monoxide. For example, hemoglobin is allowed to bind O_2_ when the oxygen concentration gets almost saturated, which commonly happens in the lungs. Then the oxyhemoglobin goes around the body, transported by the bloodstream. With O_2_ concentration decreasing, hemoglobin delivers O_2_ to myoglobin, which plays a key role to store oxygen in respiring tissues.

All the four subunits in hemoglobin correspond to either the tense (T) or relaxed (R) state simultaneously. Both the two structures have binding abilities for ligands, while the affinity differs between the T and R structures [20]. The oxygenation happens along with the rotation of the interface between α_1_β_1_ and α_2_β_2_ dimers, movement of heme Fe (II) into the porphyrin plane, breaking of the salt bridges between subunits, and formation of new interactions among residues in the α_1_β_2_ and α_2_β_1_ interfaces, which accomplished the transition from T structure to R structure (Figure 3A) [21]. Along with the crystal structure of T and R state structures, Fan et al. determined the solution structures of carbonmonoxy hemoglobin (HbCO) [22]. Their results indicate that there is a great deal of flexibility in this molecule, among which the ten best solution structures were selected and imposed with the crystal structure (Figure 3B). Thus, there are versatile types of binding ways between amino acids residues in the deoxy and oxy (or carbonmonoxy) forms, which leads to the allosteric effects on the structural, dynamic, and functional properties of hemoglobin. So far, hemoglobin has been a fruitful system to investigate further structure-function relationships and allostery in proteins.

Like hemoglobin, myoglobin is also endowed with allostery, which is flexible to bind various anions such as CN^−^, N_3_^−^, and NO_2_^−^. Comparably, the interactions between heme proteins with O_2_ and CO are most attractive [23]. As to the main biological role that myoglobin plays in the body, it reversibly binds oxygen to transport oxygen from the sarcolemma to the mitochondria of the muscle and combines excess nitric oxide to improve the aerobic metabolism of mitochondria. To confirm the binding mechanism, a great number of mutant proteins have been prepared to characterize the interaction between myoglobin and O_2_ or CO (Figure 2A,B), which provide powerful evidence that myoglobin has a specific cognitive ability for the two similar molecules.

### 2.4. Other Heme Proteins

There are other heme proteins such as cytochromes, catalases, and some peroxidases from daily food intake, at a lower level than hemoglobin and myoglobin. For example, cytochrome is a kind of electron transfer protein with heme as an auxiliary group which is widely involved in the redox reaction of animals, plants, yeast, aerobic bacteria, and anaerobic photosynthetic bacteria. As an electron carrier, cytochrome transfers electrons through the reversible change between the reduced state (II) and the oxidized state (III) of the iron atom of its heme auxiliary group. According to the different structures of heme cofactors, cytochrome can be divided into a, b, c, and d. With different physical properties of redox potential, mode of heme attachment, ligands, spin state, and degree of heme exposure [24], cytochromes play an important role in electron transferring and act as a significant component of the membrane.

In recent research, more detailed information about cytochromes is revealed, as a clearer mechanism in enzymatic reaction on the cell membrane [25], evolutionary relationship [26], and interactions between different cytochromes [27]. Based on the existing function mechanism of cytochromes, researchers got to assume the function of the related enzymes, such as Cb5R (cytochrome b5 reductase), which was found one of the main apoptosis-defensive components during cell death by stimulating NADH consumption. Although in relatively lower amounts in cells, these heme proteins have an important effect on the redox reactions in the biological membranes in various parts of organisms. More interestingly, Tezcan and co-workers conducted a great number of protein scaffolds using cytochrome, owing to its allosteric flexibility. Redesign of the protein interfaces leads to the structural and functional changes of cytochrome, which realized the metal-controlled assembly [28,29].

## 3. Applications in the Food Industry

### 3.1. Iron Supplements

As mentioned above, heme proteins and their derivatives are efficient iron sources, which are rich in food with higher bioavailability which means that they would take an active part in iron supplements. At present, the main iron supplements are divalent non-heme iron, including ferrous sulfate, ferrous gluconate, etc. However, these small molecular supplements have some side effects. Fe (II) can induce the formation of free radicals, which can exert toxic effects on cells. Moreover, the iron element in this process is easily interfered with by the chelators such as phytic acid and tannin in food which reduces the iron absorption efficiency and affects the iron supplement effect. Therefore, it is imperative to improve the bioavailability of iron and develop efficient and safe iron supplements.

Natural proteins are getting more and more popular as nutritional supplements, owing to their advantages such as high compatibility and low toxicity. Ubiquitously existing in various foodstuffs, food-derived iron-containing proteins are the main source of essential iron nutrients, which are used by living cells for growth, and to build proteins, hormones, and enzymes that regulate chemical reactions in the body. Compared to the synthesized ferrous salts, it has been shown that iron-containing proteins from nature play an important role in the regulation of iron metabolism balance and have the function of relieving the toxicity of Fe (II) [30]. Due to the coexistence with protein, iron will not be in direct contact with the chelating agents in food, thus improving the bioavailability of iron. It was also proved that heme can improve the absorption of non-heme iron in the human intestine [31]. Therefore, iron-containing proteins have a biological prospect as an efficient iron supplement, especially bovine and porcine hemoglobin.

As shown in Table 1, bovine and porcine hemoglobin, and the hydrolyzed peptides have been widely applied as stable, readily available iron supplements with higher content in food and fewer side effects. In Tang’s research [32], compared with the control group, the anemic group without treatment, and anemic groups treated with FeSO_4_, heme iron-enriched peptide had effective restorative action returning the serum iron content and Fe absorption. Furthermore, the peptides also exhibited strongly in vitro and in vivo antioxidant activities, which may be exploited as a safe, efficient new iron supplement. More interestingly, they prepared the heme iron from industrial by-product, the bovine blood, and prove that the heme iron without protein or peptides also possesses great antioxidant and anti-anemia activity [33].

### 3.2. Coloring & Pigments

Heme proteins also have a significant role in the food industry, especially as the main component of red pigment that has a close relationship with meat color [39]. It is indicated that the oxidation state of myoglobin influences the meat color during refrigeration. By Raman spectra, it is determined that with the increase of storage time, the change of secondary structure leads to following molecular rearrangement, along with the disulfide bonds changed, ultimately causing the decrease or even loss of the specific function of myoglobin as a pigment-protein [17]. Based on the characteristic analysis, Zhang and co-workers found that the interaction between myoglobin and mitochondria positively affects meat quality, in which mitochondria can promote the reduction of methemoglobin in browned meat and improve the color. Meanwhile, myoglobin can promote aerobic metabolism of muscles, transfer fermented muscle fibers to oxidized ones, and improve meat quality [46]. Due to the coloring effect, heme proteins, such as porcine hemoglobin, are always added to meat products for stable color development and less usage of artificial food ingredients such as nitrite. But according to the results of He’s group, gallic acid and tea polyphenols exhibit tremendous effects to maintain the color stability by preventing oxidation of hemoglobin [47,48]. Chhem-Kieth and co-workers [49] established a novel method to encapsulate hemoglobin pigment by four ligands, including sodium nitrite, 4-methylimidazole, methyl nicotinate, and pyrazine. It is proved that the redness of the heme-pyrazine complex showed improvement upon heating. All four complexes exhibited better stability during long-time storage.

### 3.3. Antibacterial Effects

Based on hemoglobin’s characteristic of combining reversibly with iron, it was discovered recently that interactions of hemoglobin with IsdB (Iron-Regulated Surface Determinant) system of a multidrug-resistant bacteria, Staphylococcus aureus, shed light on a rational method of antibacterial molecule discovery [50]. The bacteria Staphylococcus aureus, regarded as one of the most threatening methicillin-resistant pathogens, obtains access to organic iron mainly through the mechanism of expressing hemoglobin receptors. As specialized oxygen carriers, encapsulated hemoglobin carried in vesicles has been used in multiple studies to mimic red blood cells, which does not affect its oxygen binding properties, and oxygen off-loading can be regulated by membrane thickness of appropriate block copolymers to modulate O_2_ release and avoid side effects [51].

In addition, by fabricating enzymatic cascade microreactors coupling glucose oxidase and hemoglobin, which was proved to effectively produce hydroxyl radicals at a lower pH, thus inhibit bacterial growth and biofilm formation [52]. This mechanism was also found efficient in combating the methicillin-resistant S. aureus in the presence of glucose. Moreover, researchers found that a peptide termed HBB, which is purified from a 36-mer derived from the C-terminal region of the hemoglobin β subunit and abundant in the placenta, was regarded as a broad-spectrum antibacterial agent previously and was assumed an anti-Herpes-Simplex-Virus 2 (HSV-2) mechanism [53]. Peptides hydrolyzed from bovine and porcine hemoglobin are proved lasting and broad-spectrum antibacterial. The Hb-derived peptides also exhibit preferable antibacterial, antifungal, and anti-yeast activities at a lower concentration. In Sanchez-Reinoso’s research [41], both bovine and porcine hydrolysates produced at pH 2 and pH 3 showed antibacterial activity against L. ivanovii and anti-yeast activity against R. mucilaginosa. Conversely, only antifungal activity against M. racemosus in the porcine hydrolysate at pH 3 was observed.

### 3.4. Medical & Nutritional Applications

Due to the ability to combine with oxygen reversibly, heme proteins have been applied in the medical field. As myoglobin facilitates the diffusion of oxygen to the mitochondria in vivo, it was later determined that it could provide scalable and sustainable protein resources in the applications of oxygen therapeutics, a series of protein medications being developed as an alternative to donated blood in clinical situations. Greatly promoted using recombinant technology, studies on oxygen therapeutics are setting the technology free from the reliance on heme proteins from animal sources or excess donated blood and creating more possibilities for potential pharmaceutics [54].

Additionally, an important component in plasma, the changes caused by myoglobin in the atmospheric pressure plasma (APP) technology, a novel non-thermal decontamination technology has attracted our attention. To solve the problem of APP-induced green discoloration of myoglobin, Yong and colleagues have developed a control measure to add reducing agents. Lower concentration of reducing agents, such as 0.1% sodium dithionite, could prevent discoloration effectively, while a higher level (5%) would induce red coloration. The detailed mechanism would help promote the application of APP in food industry, decrease foodborne illness, and improve food quality [55].

With the furthering of researches, the good solubility, emulsification, and foaming properties of hemoglobin make it more widely used in food. Hemoglobin derivatives, moreover, will enhance the inherent properties of hemoglobin while giving the product new functions. Adding other functional heme proteins, such as by-products of hemoglobin, to meat products is proved to promote the emulsifying capacity and stability prominently as well as to increase the iron content. Thus, heme proteins were assumed an alternative food addictive to sodium salts or casein [40,56,57,58]. As shown in the Table 1, hemoglobin and its peptides can form different types of emulsions, which are proved better emulsifiers in food, providing heme iron.

Peptides derived from regions of heme proteins are also determined efficient taste attributes. Hydrolyzed by different enzymes, these peptides exhibit a better taste, aroma development, and overall sensory enhancement, as well as less metallic taste caused by heme iron [43,44,45]. However, with light sensitivity, decreased stability, the undesirable dark color, and an unpopular smell, more practical use of hemoglobin and its derivatives is still limited. More methods of preprocessing and ingredients are to be found for better stability and the oxidation state of the heme iron, which correlate closely with the properties above.

## 4. Discussion

### 4.1. Denaturation and Oxidation

The denaturation and oxidation of heme-binding proteins are tightly connected, which may affect the color, taste, and texture properties of heme containing food. The heme-iron atom can undergo autoxidation and convert from the ferrous to the ferric state. Brantley et al. [58] proved the pathway of heme oxidation, in which two possible mechanisms were summarized (Figure 4). The first intermediate, Fe^2+^O_2_ complex, also acted as the origin of the two ways. Firstly, O_2_ dissociates from the heme, leading the ferrous iron of the deoxy-protein to interact with water. Following the conversion from O_2_ to O_2_^−^, Fe (II) is oxidized to Fe (III). On the other hand, oxidation is realized by the dissociation of HO_2_. Being unable to bind oxygen, the protein with oxidized iron atoms is inactivated, which would result in increasing hemin loss and denaturation of the globin eventually. To solve the problem of denaturation, it has been indicated that the zwitterionic detergent Empigen BB (EBB, N, *N*-Dimethyl-*N*-dodecylglycine betaine) was not able to induce a total denaturation, but can lead to the dissociation of the heme group of the myoglobin [59].

Moreover, food-derived myoglobin is an important nutritional source of bioavailable iron. Though the heme-iron, such as myoglobin, takes a smaller part in non-vegetarian diets, it is more easily absorbed by the body and has a higher biological activity than non-heme iron.

### 4.2. Absorption Rate and Bioavailability

There are still many questions regarding the applications of heme proteins in dietary supplements, such as the relationship between digestion and absorption of these proteins; the structure and nutritional function of iron-containing proteins denatured in food processing; etc. The complex interactions between dietary heme proteins, various animal-sourced and plant-sourced food, and other food-borne components might lead to a decreased absorption rate or a lower bioavailability sometimes.

The bioavailability of heme proteins is further studied nowadays. By measuring the absorption rate of heme proteins along with other food-borne compounds, researchers suggested that the human uptake is slightly affected by soy proteins, while other cereals and legume proteins do not have a notable effect [60]. It was also determined in vitro that some dietary proteins, such as animal-based collagen and vegetable-based zein and glutelin, could promote the absorption of heme proteins after purified (e.g., hemoglobin), which would lead to a decrease, in contrast, if unpurified [61]. Additionally, some proteins would also reduce heme iron availability such as casein from milk, and gliadin from rice [61,62].

It was assumed that the uptake mechanism might relate to interactions between various dietary compounds and certain receptor for heme, as well as different sources for food proteins [61]. Additionally, it was assumed that the effect was caused by concentration of heme iron, as globin (heme-removed hemoglobin) could improve iron uptake by combining with heme [63,64]. In a similar method, uptake of hemoglobin along with red blood cell concentrate or with additional beef would significantly promote the bioavailability of heme iron [32].

### 4.3. Nutritional Problems

Apart from the taste and smell of heme proteins, it is assumed that heme iron might have some nutritional problems for human health. It is acknowledged that heme protein from red meat is an important source of dietary iron intake for its abundance as well as bioavailability. However, excessive consumption of red meat, its products, and heme iron, especially in developed countries, can increase the risk of various diseases [65]. A great number of research proved that heme iron might have adverse effects on cancer pathogenesis through the catalysis of N-nitroso compounds (NOC) [66], the lipid oxidation and thereby formation of cytotoxic and genotoxic aldehydes [67,68], and oxidation caused by overhigh iron levels in vivo [69].

To balance the nutritional value and underlying problems brought by red meat, regulating the reference intake is a promising method. According to the World Cancer Research Fund, consumption of red meat of less than 500 g per week and less processed meat were recommended [70]. Moreover, researchers suggested adding antioxidant ingredients, such as plant extracts etc., [71] and proper ways of packaging could be possible methods to cut down detrimental effects of heme iron and improve the quality of meat [65].

## 5. Conclusions

This review summarized recent advances in the structure, function, nutrition, and applications of several typical heme proteins naturally occurring in foods, which hold great promise for iron supplementation. Heme proteins, including hemoglobin and myoglobin, are well developed in food processing, owing to their high content in meat and its by-product. Besides the proteins described here, there are still some other heme proteins neglected in this review, owing to their similar structure to the above proteins or their minor content in foodstuffs. Currently, the structure and function of heme proteins have been deeply evaluated by scientists with the development of physical and biological technology.

Despite the progress, there have been some questions/issues that remain to be answered/addressed in the future studies. First, more practices will be needed to invest in the usage of iron-containing proteins, performing as a safe and regulatory iron supplement. Second, better evaluation systems should be established to assess the absorption efficiency of heme iron in a real, complicated food system. Third, the effect of other food components on the structure and iron supplement function of heme protein is required for elucidation. Finally, the bioavailability of entire foodstuff containing heme protein, which is a major concern during food intake, has not yet been elucidated.

## Figures and Tables

**Figure 1 foods-11-03594-f001:**
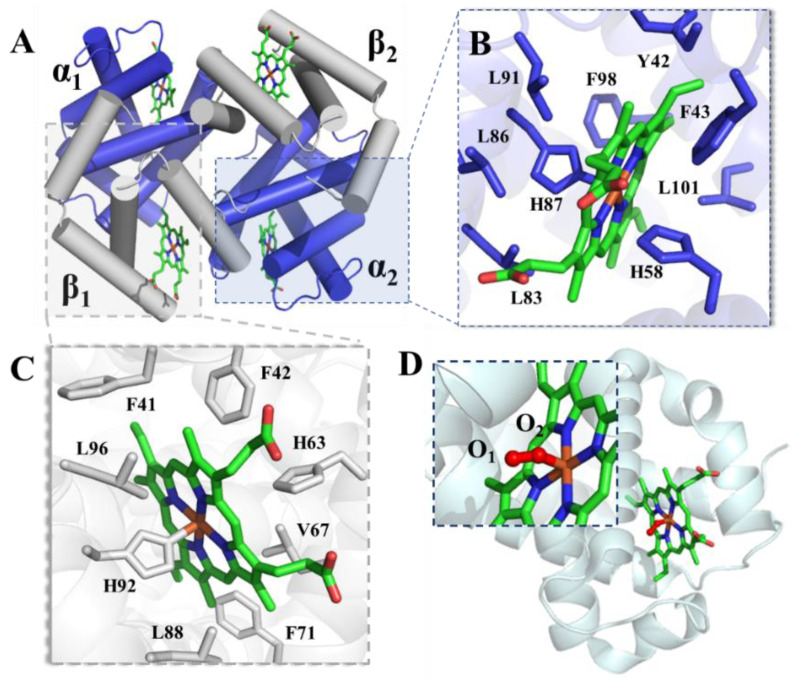
(**A**) Three-dimensional structure of hemoglobin from Bos taurus (PDB number: 6ii1) [8], composed of two α chains and two β chains. α chains are presented in blue (**B**), and β chains are presented in gray (**C**). (**D**) Oxygenated state of goose hemoglobin (PDB number: 1faw) [9].

**Figure 2 foods-11-03594-f002:**
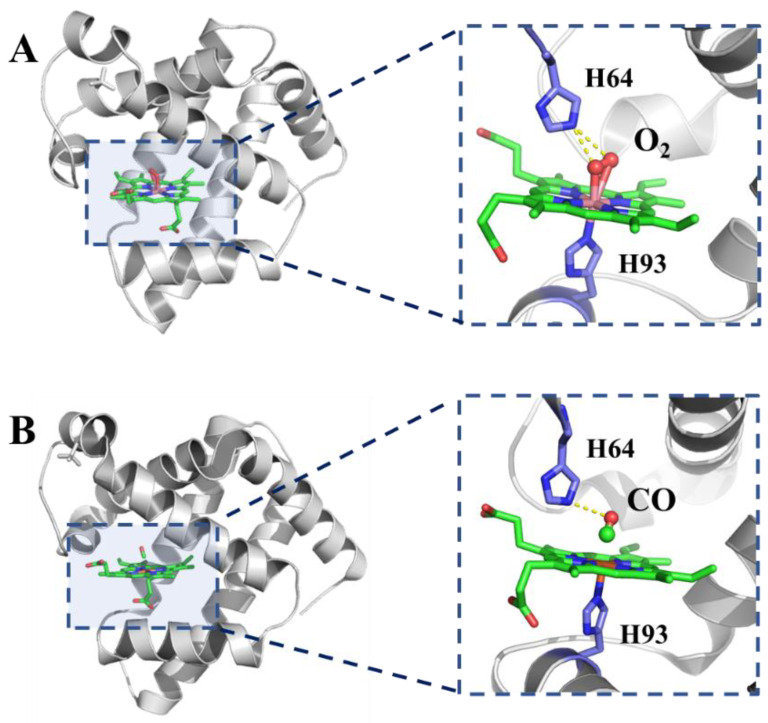
Crystal structure of myoglobin binding O_2_ ((**A**), from whale, PDB number: 1yoi) [14] and CO ((**B**), from horse, PDB number: 1dwr) [15].

**Figure 3 foods-11-03594-f003:**
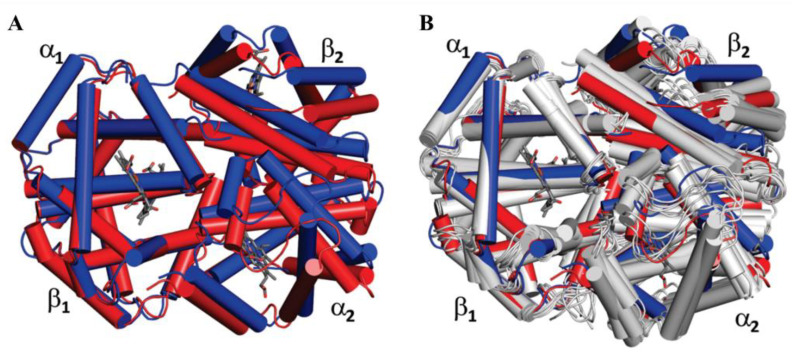
(**A**) Superimposed comparison between the T (PDB number: 4hhb, blue) and R (PDB number: 2dn3, red) state structures of hemoglobin. (**B**) The ten lowest energy solution structures of HbCO obtained by NMR (PDB number: 2m6z, gray) are superimposed with the T and R structures of human adult hemoglobin (Hb A) obtained by X-ray crystallography [21]. Reprinted/adapted with permission from Ref. [21]. 2015, Yuan et al.

**Figure 4 foods-11-03594-f004:**
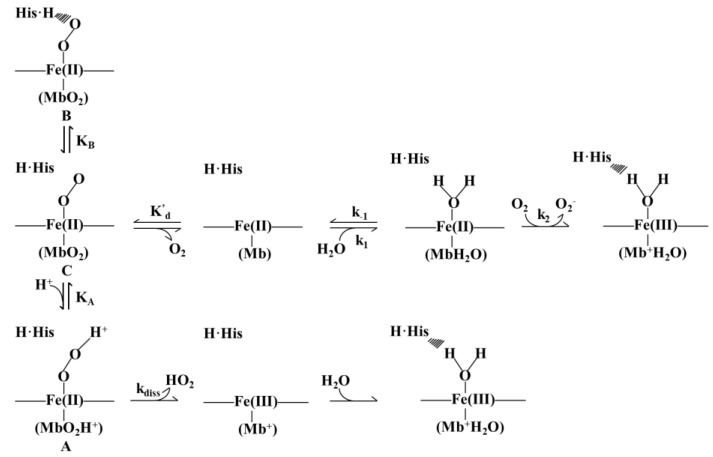
Schematic presentation of reactions in converting the heme-iron atom from the ferrous (Fe^2+^) to the ferric (Fe^3+^) state.

**Table 1 foods-11-03594-t001:** Applications of heme proteins in food areas.

Applications	Proteins	Advantages	**References**
Iron supplements	Hb-B	Readily available; less gastrointestinal discomfort	[34]
	Hb-B & Hb-P	Good storage stability	[35]
	Hb-P	Higher content and good stability	[36]
	Heme-enriched peptide of Hb-B	Higher iron bioavailability and fewer side effects	[32]
Coloring & pigments	Arg-Hb	Color stability during process	[37]
	COHb-P	Significant increase in pigments	[38]
	Hb-P	Storage and thermal stability; reduction of nitrite addition	[39]
	Arg-Hb	Natural resource and color development	[40]
Antibacterial & antioxidant	Peptides of Hb-B & Hb-P	Antibacterial, antifungal, and anti-yeast activities	[41]
	Peptides of Hb-P	Better antioxidant properties at low concentrations; lasting	[42]
Emulsifiers	Hb-P	Forming o/w emulsions	[38]
	Hb-P & Peptides of Hb-P	Better emulsifiers to plant proteins; superior to casein	[40]
		Forming both o/w and w/o emulsions	[42]
Taste attributes	Peptides of Hb-P (exopeptidase hydrolysis)	Less bitter and higher umami taste	[43]
	Peptides of Hb-P (papain hydrolysis)	Aroma development	[44]
	Both Hb-P and the peptides (γ-GT hydrolysis)	Overall sensory enhancement	[45]

Abbr: GA—Gallic Acid/GT—Glutamyltranspeptidase/Hb-B—Bovine Hemoglobin/Hb-P—Porcine Hemoglobin/o/w—oil in water/w/o—water in oil.

## Data Availability

The data presented in this study are available on request from the corresponding author.

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
