# Peer review of "Dietary Heme-Containing Proteins: Structures, Applications, and Challenges"

_foods, 2022, doi:10.3390/foods11223594_

Round 1

Reviewer 1 Report

The manuscript “Dietary Heme-containing Proteins: Structures, Applications, 2 and Challenges” is an interesting review. In this review, the authors summarized the structural and functional properties of hemoglobin, myoglobin, and other heme proteins and highlighted recent advances in applications of heme proteins as dietary iron sources in the field of food science and nutrition. While I believe this topic is of great interest to readers, I think some revision is required. So, I recommend this manuscript for publication with minor revisions.

1.    In Section 2.2 and 2.3, figure numbers are incorrect.
Line 65: Fig. 2A -> Fig. 1A
Line 69: Fig. 2B, C -> Fig. 1B, C
Lien 72: Fig. 2D -> Fig. 1D
Line 100: Fig. 3 -> Fig. 2
Line 110: Fig. 3A -> Fig. 2A
Line 111: Fig. 3 -> Fig. 2

2.    The notation for figures should be unified (Fig. or Figure). I think that it is better to unify that for hemoglobin (most of the notation in the text is "hemoglobin," but "Hb" is also used in some places, e.g., lines 73, 109, and 249).

3.    For Fig. 1, the description for heme and the biological source for A are required.

4.    For Fig. 1, I think that it is better to use a structure of reduced hemoglobin for A-C because the authors mentioned the structural difference between the reduced and oxygenated hemoglobin.

5.    For Fig. 1D, I think "oxygenated state" is correct, not "oxidized state."

6.    Although the allostery in hemoglobin is discussed in Section 2.2, the authors described that the whole structure maintains almost no change compared with reduced Hb in lines 72-73. I think the description here should be revised because structural changes are also occurring within each subunit upon the oxygenation.

7.    In Line 110, the authors mentioned function of the lateral phenylalanine. I think it is better to represent the location of this residue in Fig. 2A.

8.    Figure 3 should be moved into Section 2.3.

9.    For Fig. 3B, the description of "with the T and R structures of Hb A ..." should be corrected.

10. The abbreviation of myoglobin should be defined.

11. The references for the studies of myoglobin mutants (lines 121-124) are required.

12. In lines 129-130, the authors described "the allosteric effect refers to the phenomenon that multi-subunit proteins change their conformation and activity...," but the allosteric effect is observed also in the monomer protein.

13. I couldn't understand the description in lines 143-144: "except for the crystal structure of T and R state structures, ...."

14. The sentence in lines 153-154 should be corrected: While it prefers to bind O2, CO, and NO in its reduced state.

15. For line 155 and line 313, O2 -> O2.

16. For line 172, I think the present tense "act" is correct, not "acted."

17. I think the authors should revise the statements in lines 192-195. "On the other hand," when used in succession, is difficult to understand.

18. Figure 4 is better to be made in high resolution.

19. I think the authors should revise the sentence in lines 233-238. It is very long, and I'm not sure what "for" in the line 235 means grammatically.

20. For line 259, I don't think it is necessary to redefine abbreviation of hemoglobin.

21. For line 265, I think the non-abbreviated name for HSV-2 is required.

22. I think the authors should revise the sentence in lines 285-286.

23. I think the authors should revise the sentence in lines 286-288.

24. For line 295, does "the table" mean Table 1?

25. I think the section title of 3.4 should be more specific.

26. I think the authors should revise the sentence in lines 312-313.

27. I think the authors should revise the sentence in lines 316-318.

28. I think the authors should revise the sentence in lines 343-345.

29. There are abbreviations that are not used in subsequent sentences (e.g. GOx-Hb MRs, MRSA, and RBCC).

30. In Reference section, "1." at the beginning of the line 499 (reference 53) should be deleted.

31. Some parts of the manuscript are difficult to read, and there are some grammatical mistakes. Please improve the English of the manuscript.

Reviewer 2 Report

Xing et al present a review on heme proteins, entitled " Dietary Heme-containing Proteins: Structures, Applications, and Challenges ". The topic is, in my opinion, interesting and innovative. However, the manuscript presents some weaknesses.

The different ideas are not properly linked. For instance, I'm not sure that meat and blood can be considered as promising dietary sources for iron supplementation, but rather as elements that should be included in a balanced alimentation. The design of the manuscript has to be improved : L 51 : 2 structure of heme proteins, two examples, then allosteric effect thean other proteins?

The style is poor and sometimes unprecise, and in my opinion, leads to some misunderstandings. For instance, L27, anemia is the most severe stage of "its" presence, what is its? L30, what is the relation between iron bioavailability and iron deficiency worldwide? L54 why do the authors focus on livestock, on hemoglobin? L58 why do the authors talk about centrifugation, mixing both circulating blood and meals? L206: Researches show...is not a sentence for a scientific journal. L188: typing error...

Regarding the figures, the reader is unable to know where the pictures come from since the origin (publication) is not clearly mentioned. There is an important mix between species, and the choice of the authors is difficult to follow. 

As sum, the manuscript is in my opinion, too preliminary, and needs to be rewritten, including English editing, rationalization, and proper citation of sources, to meet the publication criteria of the journal.

Round 2

Reviewer 2 Report

Xing et al have taken in consideration some of the improvement suggestions i had previously made. However, the english style could still be improved in my opinion, and the following paragraphs have to be improved for clarification purpose.

L354 : which element from fish, chicken beef? Why heme only?

However, the bioavailability is decreased when the heme only was combined with fish or chicken respectively, aside from beef, which exerts no effect on it.

L357 : from which species?

Nor do purified proteins such as collagen, casein, and albumin affect the bioa- vailability of heme proteins. 

 L367 : sentence to be rewritten

Through notably  the following mechanisms, such as the catalysis of N-nitroso compounds (NOC) [70], the lipid oxidation and formation of cytotoxic and genotoxic aldehydes [71–72], and oxidation caused by overhigh iron levels in vivo [73].

Author Response

L354: which element from fish, chicken beef? Why heme only?

However, the bioavailability is decreased when the heme only was combined with fish or chicken respectively, aside from beef, which exerts no effect on it.

Response: Suggestions were followed in part 4.2. This paragraph is majored in the relationship between heme uptake and other compounds in food. To make it more clearer, we rewrote this paragraph. According to the related references, concentration of heme iron, red blood cell, and the existence of some other proteins may increase the absorption of heme.

L357: from which species?

Nor do purified proteins such as collagen, casein, and albumin affect the bioavailability of heme proteins. 

Response: Suggestions were followed in part 4.2. This paragraph has been rewritten.

 L367: sentence to be rewritten

Through notably the following mechanisms, such as the catalysis of N-nitroso compounds (NOC) [70], the lipid oxidation and formation of cytotoxic and genotoxic aldehydes [71–72], and oxidation caused by overhigh iron levels in vivo [73].

Response: Suggestions were followed in line 359.